# Screening the Key Region of Sunlight Regulating the Flavonoid Profiles of Young Shoots in Tea Plants (*Camellia sinensis* L.) Based on a Field Experiment

**DOI:** 10.3390/molecules26237158

**Published:** 2021-11-26

**Authors:** Jing Jin, Yi-Qing Lv, Wei-Zhong He, Da Li, Ying Ye, Zai-Fa Shu, Jing-Na Shao, Jia-Hao Zhou, Ding-Mi Chen, Qing-Sheng Li, Jian-Hui Ye

**Affiliations:** 1Zhejiang Agricultural Technical Extension Center, 29 Fengqi East Road, Hangzhou 310020, China; zdcxjj@126.com; 2Tea Research Institute, Zhejiang University, Hangzhou 310013, China; yiqinglv@zju.edu.cn (Y.-Q.L.); yingye@zju.edu.cn (Y.Y.); zhoujiahao@zju.edu.cn (J.-H.Z.); 3170100462@zju.edu.cn (D.-M.C.); 3Lishui Institute of Agriculture and Forestry Sciences, Lishui 323000, China; jnhwz@126.com (W.-Z.H.); shuzaifa@163.com (Z.-F.S.); snana2008@163.com (J.-N.S.); 4Institute of Sericulture and Tea, Zhejiang Academy of Agricultural Sciences, Hangzhou 310021, China; lida@zaas.ac.cn

**Keywords:** *Camellia sinensis*, light intensity, light spectral composition, catechins, flavonol glycosides, shade treatment, co-expression

## Abstract

Both UV and blue light have been reported to regulate the biosynthesis of flavonoids in tea plants; however, the respective contributions of the corresponding regions of sunlight are unclear. Additionally, different tea cultivars may respond differently to altered light conditions. We investigated the responses of different cultivars (‘Longjing 43’, ‘Zhongming 192’, ‘Wanghai 1’, ‘Jingning 1’ and ‘Zhonghuang 2’) to the shade treatments (black and colored nets) regarding the biosynthesis of flavonoids. For all cultivars, flavonol glycosides showed higher sensitivity to light conditions compared with catechins. The levels of total flavonol glycosides in the young shoots of different tea cultivars decreased with the shade percentages of polyethylene nets increasing from 70% to 95%. Myricetin glycosides and quercetin glycosides were more sensitive to light conditions than kaempferol glycosides. The principal component analysis (PCA) result indicated that shade treatment greatly impacted the profiles of flavonoids in different tea samples based on the cultivar characteristics. UV is the crucial region of sunlight enhancing flavonol glycoside biosynthesis in tea shoots, which is also slight impacted by light quality according to the results of the weighted correlation network analysis (WGCNA). This study clarified the contributions of different wavelength regions of sunlight in a field experiment, providing a potential direction for slightly bitter and astringent tea cultivar breeding and instructive guidance for practical field production of premium teas based on light regimes.

## 1. Introduction

Tea is a popular beverage with a unique taste, which is produced from the leaves of *Camellia sinensis* (L.) O. Kuntze. Tea is associated with numerous health benefits, such as antitumour, cardioprotective, and antidiabetic effects [1,2,3]. Flavonoids contribute to the taste of tea and also contribute to the bioactivities of the tea leaves. Flavanols and flavonols are the top two flavonoid subclasses in tea leaves [4]. The predominant flavanols in fresh tea leaves are catechins, accounting for 70~80% of tea polyphenols. (−)-Epigallocatechin gallate (EGCG), (−)-epicatechin gallate (ECG), (−)-epigallocatechin (EGC), (−)-epicatechin (EC), (+)-gallocatechin (GC) and (+)-catechin (C) are the major catechin compounds in fresh tea leaves. Second to flavanols, flavonols account for ~13% of tea polyphenols [5], mainly consisting of the *O*-glycosides of quercetin, kaempferol and myricetin [6]. Although the contents of flavonols and their glycosyl derivatives are much lower than catechins, flavonol glycosides are also important bitter and astringent compounds in tea infusions, considering their extremely low thresholds and ability enhance the bitterness of caffeine [7]. Moreover, the biosynthesis of flavonoids in plants is associated with their adaptability to the environment [8,9] or responses to environmental stimuli, such as light and plant growth regulators and drought stresses [9]. The accumulation of flavone glycosides in the leaves of *Plectranthus scutellarioides* has been shown to be affected by light condition [10,11]. The biosynthesis of flavonoids in regular tea cultivars was shown to be attenuated by low illumination [12]. Thus, black net shade treatment is a common agronomic practice for producing high-grade green tea in several countries, such as China and Japan, due to its reduction effect on light intensity. 

Sunlight consists of UV light (below 400 nm), visible light (400–495 nm of blue light, 495–570 nm of green light, 570–590 nm of yellow light and 590–710 nm of red light) and infrared radiation (710–1000 nm) [13]. Light regulates the growth of plants as a signal [14], and different wavelength regions of sunlight may play different roles in plant physiology [15]. Photoreceptors of blue light, red light and UV B have been discovered in plants, implying the important influence of the wavelengths on the growth of plants [13]. The promotion of UV leading to the accumulation of flavonol glycosides in tea plants was confirmed by covering tea plants with special UV-B-excluding screens [16]. In recent years, blue and red light-emitting diodes (LED) have been used to alter light quality, including use in greenhouses to improve the growing conditions of plants [17]. The regulatory effect of blue LEDs on flavonoid biosynthesis in the young shoots of tea plants has been reported [18,19]. However, most studies have been carried out under artificial light conditions, and the respective contributions of UV, blue and red light regions of sunlight to the biosynthesis of flavonoids are still unclear. Understanding the key region of sunlight regulating the biosynthesis of flavonoids is important for premium tea cultivar breeding based on biotechnology, as well as for development of relevant agronomic operations. Additionally, the LED facilities are difficult to implement for wide application in the field due to the limited intensity of LEDs, the supply of electric power and the high costs for long-term service and maintenance. Although plastic films (e.g., red, yellow, green, blue and white) can also alter the light conditions of field or indoor cultivation systems [20], the inputs of plastic films and relevant facilities equipped with new irrigation systems are relatively high, which may not be suitable for woody plants such as tea plants. A field applicable method for altering the light conditions is desirable.

Here, we investigated the impacts of different shade treatments, including black nets with different shade percentages and blue, yellow and red nets, on the flavonoid compositions in the young shoots of tea cultivars (‘Longjing 43’, ‘Zhongming 192’, ‘Wanghai 1’, ‘Jingning 1’ and ‘Zhonghuang 2’). Principal component analysis (PCA) was employed to screen the key factor regulating the profiles of flavonoid compounds in tea leaves. A weighted correlation network analysis (WGCNA) was used to explore the correlations between the metabolite contents of tea samples and the parameters of light conditions altered by agronomic practices, so as to reveal the key region of sunlight regulating flavonoid biosynthesis in tea shoots. 

## 2. Results

### 2.1. Alteration Effects of Different Shade Treatments on the Light Condition Underneath

Table 1 lists the basic light condition parameters under different polyethylene shade nets. The light spectrograms under different shade treatments are shown in Figure 1. The light intensity and UV intensity data were obtained from our previously published study, which were assessed together [21]. For black shade nets, both light intensity and UV intensity were greatly reduced as the shade percentage increased from 70% to 95%, whereas the light spectral composition was hardly affected. For colored shade nets, shade treatments not only reduced the light and UV intensities but also altered the light spectral composition. Specifically, the blue net shade treatment largely increased the blue and green ratios (%) in the light spectrum from 31.7% and 25.8% (natural sunlight) up to 50.4% and 34.5%, respectively, whereas it reduced the yellow and red ratios. The yellow net shade treatment elevated the yellow and red ratios in the light spectrum but reduced the blue and green ratios. The highest ratio of red light (61.0%) was achieved under the red net shade treatment, while the ratios of blue, green and yellow light were relatively low. Thus, black net shade treatments reduced the light and UV intensities without impacting the light spectral composition, while colored net shade treatments not only reduced the light and UV intensities but also largely increased the ratio of light corresponding to the net color. 

### 2.2. Impacts of Shade Treatments on the Profiles of Catechins in the Young Shoots of Different Tea Cultivars

Five tea cultivars, including ‘Longjing 43’, ‘Zhongming 192’, ‘Wanghai 1’, ‘Jingning 1’ (temperature-sensitive albino tea) and ‘Zhonghuang 2’ (light-sensitive albino tea), were used to investigate the cultivar-dependent responses to shade treatments. Different shade treatments were performed according to the same method as reported previously, using an open all-around shade mode to achieve consistent air temperature and humidity [21]. The tea plants of each cultivar were shaded using black and colored polyethylene nets, with tea plants grown in natural sunlight used as control (CK). The young shoots were respectively harvested at the standard of two tea leaves and one bud on the 5th, 10th, 15th, 20th and 25th days of shade treatments and immediately submitted to fixation and drying to preserve the authentic chemical composition of the tea leaves. The tea samples were respectively termed BN70%_No. 1–5 (black net with the shade percentage of 70%), BN95%_No. 1–5 (black net with the shade percentage of 95%), BN_No. 1–5 (blue net), YN_No. 1–5 (yellow net) and RN_No. 1–5 (red net) based on the shade treatments, followed by the serial number indicating the batches of tea samples harvested in order. Appendix A shows the contents of catechins in the tea samples of different cultivars under shade treatments, while the dynamic changes in total catechins (TC, the sum of 8 catechin compounds) are shown in Figure 2. For the tea trees grown in natural sunlight, CK_No. 1 of ‘Jingning 1’ contained the highest content of TC (231.10 mg dry weight (DW)/g), subsequently followed by ‘Zhongming 192’ (203.08 mg DW/g), ‘Wanghai 1’ (194.07 mg DW/g) and ‘Longjing 43’ (165.31 mg DW/g), while ‘Zhonghuang 2’ contained the lowest content of TC (156.67 mg DW/g). During the observation period (up to 25 d), the contents of TC in the CK samples of ‘Zhongming 192’ and ‘Jingning 1’ were relatively stable (<12.3% of percentage change), while greater variations (17.2%~18.6%) in TC content was observed for ‘Longjing 43’, ‘Wanghai 1’ and ‘Zhonghuang 2’. Under different shade treatments, ‘Zhongming 192’, ‘Jingning 1’ and ‘Zhonghuang 2’ showed less sensitivity to the altered light conditions (<15.8% TC reduction compared with the corresponding CK samples), while the contents of TC in ‘Wanghai 1’ and ‘Longjing 43’ were more obviously impacted by the different shade treatments, with TC reductions of 35.7% and 24.4%, respectively. 

### 2.3. Impacts of Shade Treatments on the Profiles of Flavonol Glycosides in the Young Shoots of Different Tea Cultivars

Appendix A shows the contents of flavonol glycosides in the tea samples under different shade treatments, while the dynamic changes in total flavonol glycosides (TFG, the sum of 13 flavonol glycosides) in different tea cultivars are shown in Figure 3. CK_No. 1 of ‘Zhongming 192’ contained the highest content of TFG (7412 μg DW/g), followed by ‘Zhonghuang 2’ (5562 μg DW/g), while ‘Jingning 1’, ‘Wanghai 1’ and ‘Longjing 43’ contained relatively lower contents of TFG (~4500 μg DW/g). During the observation period, the content of TFG remained stable for ‘Zhongming 192’ and ‘Zhonghuang 2’ (<2.4% of percentage change), whereas the contents of TFG in ‘Longjing 43’, ‘Wanghai 1’ and ‘Jingning 1’ were respectively increased by 41.9%, 26.9% and 28.1% at the end of the observation period (middle of July). All shade treatments notably reduced TFG compared with TC. The percentage reductions in TFG were generally elevated with increases in the shade percentage from 70% to 95%, although tea cultivars showed differential responses to colored net shade treatments (Figure 2 and Figure 3). This suggests that light intensity and UV intensity might importantly participate in the biosynthetic regulation of flavonoid glycosides. After the black net 95% shade treatment for 20 d, 64.0% of the TFG was reduced in the BN95%_No. 4 treatment of ‘Longjing 43’, followed by ‘Zhonghuang 2’, ‘Jingning 1’ and ‘Wanghai 1’, while 42.1% of TFG was reduced for ‘Zhongming 192’. The percentage reductions of TFG were higher than those of TC, with greater cultivar-dependent differences in responses to light conditions. Moreover, notable decreases in TFG were observed in the BN95%_No. 5 samples of all the tea cultivars, implying that the shade-induced decrease in TFG could be divided into two phases, namely short-term (less than 20 d) and long-term impacts (longer than 20 d). 

Flavonol glycosides can either be classified by aglycone, namely quercetin glycosides (Q-glycosides), kaempferol glycosides (K-glycosides) and myricetin glycosides (M-glycosides); or classified by the sugar moiety, such as mono-, di- and triglycosides. Figure 4 and Figure 5 show the contents of different flavonol glycoside groups in the young shoots of tea cultivars under shade treatments. The constitution of flavonol glycosides was cultivar-dependent. Based on aglycone, Q-glycosides were the principal flavonol glycoside group in the tea samples of ‘Longjing 43’, ‘Zhongming 192’, ‘Wanghai 1’ and ‘Zhonghuang 2’, but not ‘Jingning 1’. M-glycosides were the second largest flavonol glycoside group for ‘Longjing 43’ and ‘Zhonghuang 2’, while ‘Zhongming 192’ and ‘Wanghai 1’ contained comparable levels of M-glycosides and K-glycosides. Under shade treatments, Q-glycosides and M-glycosides were greatly reduced for all tea cultivars. The highest percentage reductions of Q-glycosides and M-glycosides were observed in the BN95%_No. 5 samples of ‘Longjing 43’ (83.5% and 77.7%, respectively), ‘Wanghai 1’ (93.0% and 81.6%, respectively) and ‘Zhonghuang 2’ (70.5% and 60.3%, respectively). In contrast, the highest percentage reductions of Q-glycosides and M-glycosides were observed in the colored shade net-treated samples of ‘Zhongming 192’ and ‘Jingning 1’. The percentage reductions of K-glycosides were generally less than those of M-glycosides and Q-glycosides under shade treatments. Our results demonstrated that M-glycosides and Q-glycosides were more sensitive to light conditions compared to K-glycosides. Based on the sugar moiety, both monoglycosides and triglycosides were abundantly present in the tea samples of ‘Longjing 43’, ‘Zhongming 192’, ‘Zhonghuang 2’ and ‘Jingning 1’, but not ‘Wanghai 1’, which contained much less monoglycosides, while diglycosides were present at low levels in all tea cultivars. Under the different shade treatments, the mono-, di- and triglycosides showed overall downward trends. The impacts of colored nets shade treatments on the contents of mono-, di- and triglycosides varied by tea cultivar.

### 2.4. Screening the Key Factor of the Profiles of Flavonoids Based on PCA

The accumulation of flavonoids in tea shoots was synthetically impacted by the shade treatment duration, shade net used and tea cultivar. PCA as a multivariate statistical tool has been widely used for the classification and interpretation of influencing factors. Figure 6 shows the PCA score plots and loading plots of different tea samples based on the compositions of catechins and flavonol glycosides. The first three principal components (PC) accounted for 73.4% of the total variance (PC1 = 40.9%, PC2 = 20.4%, PC3 = 12.1%, Figure 6A). During the whole observation period, the CK samples of all cultivars were clearly separated from the shade-grown tea samples. Furthermore, the tea samples of ‘Zhongming 192’ were clustered and discriminated from other cultivars, whereas the tea samples of the other 4 cultivars were not clearly distinguished. This was in agreement with the previous result showing that ‘Zhongming 192’ contained a much higher content of TFG than the other 4 cultivars. However, these tea samples could not be well discriminated based on either the shade treatment duration or type of shade net used according to the preliminary test results. Thus, the shade treatment greatly modulated the profiles of flavonoids based on cultivar features. It is worth noting that CK samples were distributed in the positive direction of PC1 discriminated from the shade-grown samples. The loading plot showed that most flavonol glycosides, including myricetin glucoside (M-glu), myricetin galactoside (M-gal), quercetin glucosyl-rhamnosyl-glucoside (Q-glu-rha-glu), quercetin glucosyl-rhamnosyl-galactoside (Q-gal-rha-glu), quercetin rhamnosyl-glucoside (Q-glu-rha), quercetin glucoside (Q-glu), quercetin galactoside (Q-gal), monoglycosides, diglycosides, triglycosides, Q-glycosides, M-glycosides and TFG, were distributed in the positive section of PC1 (Figure 6B), suggesting that the high contents of these flavonoid compounds were responsible for discriminating CK samples from the shade-grown samples. This also demonstrated that the accumulation of K-glycosides in tea shoots was less sensitive to light conditions compared with M-glycosides and Q-glycosides.

### 2.5. Association of Flavonoid Compositions with the Light Condition Parameters

WGCNA was used to explore the correlations between flavonoid compositions and the basic light condition parameters. Shade-induced reductions of the TFG content were prevalent among the different tea cultivars. To investigate the common flavonoid biosynthesis responses of tea plants to different regions of sunlight, the flavonoid compositions of all No. 4 tea samples were used to correlate the light and UV intensities, as well as the ratios of light in the spectrum on the 20th day. There were four distinct modules achieved via WGCNA, which are indicated by different colors and shown using a dendrogram in Figure 7A, while the Pearson coefficients are shown in Figure 7B (upper). Obviously, the 21 compounds and compound groups in module 3 (Turquoise) positively correlated with the light and UV intensities (Pearson coefficient = 0.95, *p* < 0.05) but showed no correlations with the blue, green, yellow or red ratios in the light spectrum (Figure 7B). The 3 compounds and compound groups in module 2, namely kaempferol glucosyl-rhamnosyl-glucoside (K-glu-rha-glu), K-glycosides and GCG, positively correlated with the light and UV intensities (Pearson coefficient <0.52, *p* < 0.05) but negatively correlated with the red ratio in the light spectral composition assessment (Pearson coefficient = −0.48, *p* < 0.05). This indicates that the biosynthesis of K-glu-rha-glu, K-glycosides and GCG was less impacted by the light and UV intensities compared with the 21 compounds and compound groups in module 3. Figure 7C shows the correlations of flavonoid contents for module 3 with the light and UV intensities. The light intensity and UV intensity were congruously correlated with these compounds and compound groups, especially M-glu, Q-glu-rha-glu, Q-glu-rha, Q-glu, TFG, M-glycosides, Q-glycosides, diglycosides and triglycosides, with Pearson coefficients being above 0.7. This was in general agreement with the PCA results. 

## 3. Discussion

Alterations of illumination conditions can modulate the growth of plants, especially relating to secondary metabolism [13,22]. Flavonoids are important secondary metabolites in tea plants, which participate in the responses to environmental stresses [22,23]. Numerous studies have shown that strong illumination is propitious to the biosynthesis of flavonoids in tea plants [12], and that the biosynthesis of flavonoid subgroups is differentially impacted [24,25]. In the present study, greater proportions of TFG were reduced in the young shoots of ‘Longjing 43’, ‘Zhongming 192’, ‘Wanghai 1’ and ‘Jingning 1’ under all the shade treatments compared with TC, suggesting that the biosynthesis of flavonol glycosides was more sensitive to ambient light conditions. The same phenomenon was observed in the second leaves of ‘Fudingdabaicha’ after shade treatments [21]. For different flavonol glycoside groups, M-glycosides and Q-glycosides were more impacted by shade treatments, showing greater percentage reductions than K-for glycosides, while mono-, di- and triglycosides were reduced congruously. This suggests that the biosynthesis of the aglycone moiety of flavonol glycosides might be more sensitive to light conditions compared with the glycosylation process. Our previous study indicated that quercetin glycosides were greatly accumulated in the second leaves of ‘Fudingdabaicha’ under strong illumination [21]. The higher light intensity also favored the accumulation of quercetin glycosides in Kale leaves (*Brassica oleracea* var. *sabellica*) [26]. The UVB-induced high accumulation of quercetin glycosides in *Arabidopsis thaliana* is associated with their more effective photoprotection compared with kaempferol glycosides [27]. Quercetin and myricetin glycosides with more phenolic hydroxyl groups have better singlet oxygen scavenging capacity compared with the corresponding kaempferols, participating in the plant response to light stress [28]. This gives a plausible explanation for the greater variations in M-glycosides and Q-glycosides as the light conditions changed. The TFG contents of ‘Longjing 43’, ‘Wanghai 1’ and ‘Jingning 1’ samples were greatly elevated in the middle and end of July. This might be attributed to the elevated temperature and illumination as the observation was extended to the middle of July. A heat-responsive UDP–flavonoid glucosyltransferase gene UGT73A17 was characterized in tea plants, which is responsible for the biosynthesis of various flavonoid glucosides [29]. Moreover, two glycosyltransferases involved in the biosynthesis of flavonol glycosides, UGT73C6 and UGT78D1, were reported in *Arabidopsis thaliana* [30]. In contrast, the contents of TFG in the tea samples of ‘Zhongming 192’ and ‘Zhonghuang 2’ remained relatively stable during the observation period, suggesting the cultivar-dependent regulation of flavonoid biosynthesis in response to environmental conditions. 

Different tea cultivars have different secondary metabolite features [31,32,33]. Our study showed that the variations in flavonol glycosides among different tea cultivars were greater than for catechin compounds, suggesting that the compositions of flavonol glycosides could be used a chemical composition index for different tea cultivars. Fang et al. recommended the flavonol triglycoside/flavonol diglycoside ratio as an indicator for discriminating different tea cultivars [34]. The biosynthesis of flavonoids in different tea cultivars was also differentially affected by shade treatments [35]. In our study, the greatest percentage reduction of TFG was observed in the young shoots of ‘Longjing 43’, while the lowest percentage reduction of TFG was observed for ‘Jingning 1’. Thus, the selection of shading-sensitive tea cultivars would be desirable for the efficient production of matcha or the utilization of summer–autumn tea. Among these 5 tea cultivars, ‘Jingning 1’ is a temperature-sensitive albino tea cultivar, while ‘Zhonghuang 2’ is a light-sensitive albino tea cultivar. It was reported that shade treatment promoted the accumulation of catechin compounds in the young shoots of the light-sensitive tea cultivar ‘Huangjinya’ [36]. A similar phenomenon was also observed in the greening tea leaves of the albino cultivar ‘Huangjinju’ under shade treatment, which had higher contents of catechins than the albino tea leaves [37]. Differing from regular tea cultivars, the enhanced biosynthesis of flavonoids in the shade-grown albino tea leaves was attributed to the differential biosynthetic mechanism of the flavonoids [35]. However, our study showed no significant impact on the content of TC in BN95%_No. 5 of ‘Jingning 1’ as compared with control, while the content of TC in BN95%_No. 5 of ‘Zhonghuang 2’ was significantly reduced (Appendix A). Liu et al. reported that shade treatment barely impacted the TC content of the light-sensitive albino tea cultivar ‘Yujinxiang’, along with a greater reduction of quercetin and lower reduction of kaempferol [35]. A decrease in TC content was also observed in the young shoots of the albino tea cultivar ‘Yujinxiang’ under shade treatment [38]. Different tea cultivars and different growing stages of tea plants, such as plant age, cultivation method (pot culture or field culture), and leaf maturity, are non-negligible factors affecting the responses of tea plants to altered light conditions. The accumulation of flavonoids in tea leaves is closely related to leaf maturity [39].

In addition to light intensity, the light spectral composition also has an influence on the biosynthesis of flavonoids in plants [19,40]. In our study, the WGCNA result showed that the contents of most flavonol glycosides in tea samples were positively correlated with the light and UV intensities, consistent with previous studies [12,22]. As we know, the blue, green, yellow and red ratios in the light spectrum account for above 99.0% of the visible light energy, meaning the light intensity approximately equals the energy sum of blue, green, yellow and red light. However, only 1 compound in module 4, namely Q-glu-rha-rha, showed a positive correlation with the red ratio in the light spectrum, which also showed negative correlations with blue and green ratios (*p* < 0.05), suggesting that light quality also influences the biosynthesis of certain flavonoid compounds and that the impact of the visible light region on the accumulation of flavonol glycosides is much lower than of the UV region. The concordant light intensity and UV intensity behavior in module 3 was attributed to the blocking effect of the polyethylene shade net, which coordinately reduced the visible light and UV intensities reaching the surfaces of tea plants. Hence, UV was the key region of sunlight that regulated the accumulation of flavonol glycosides. The promoting effect of UV on the accumulation of several flavonol glycosides in the first developing leaves of ‘Huangkui’ was proven using special UV-B-excluding screens, while a supplement of UV B reduced the levels of catechins in tea leaves [16]. UVR8, the receptor of UV B, mediated the signal transduction pathway to regulate flavonoid biosynthesis in tea plants [41]. In addition to UV, blue, green and red light exerted supplementary effects on the biosynthesis of certain flavonoid compounds. The effects of different light quality levels (e.g., red, blue, yellow and green light) on the growth of plants have been reported [42]. Blue light not only regulates multiple processes in plants, such as photosynthesis and leaf development, but is also involved in the regulation of secondary metabolisms [43,44]. The biosynthesis of anthocyanins and catechins in the young shoots of ‘Zhonghuang 3’ was enhanced by 4 h high-intensity supplemental blue light treatment during the nighttime [18]. The biosynthesis of flavonol glycosides was impacted differentially to blue light treatment [19]. The pathway of flavonoid biosynthesis was enriched under red net shade treatment [21]. In the present study, the light-quality-related accumulation of Q-glu-rha-rha in tea shoots was prevalent among different tea cultivars based on the WGCNA results, while the possibility of other flavonoid compounds being modulated by light quality in specific cultivars should be considered due to cultivar-specificity. Moreover, much higher densities of young tea shoots were observed in all cultivars under blue net shade treatment according to the field observation, as compared with the inhibitory growth of young tea shoots under the black net 95% shade treatment. A high density of young tea shoots is positively related with the production yield of tea leaves. The blue-light-induced growth of young tea shoots could be explained by the significantly elevated auxin levels in the tea samples under blue net shade treatment in our previous study [21]. The regulatory effect of blue net shade treatment on the growing conditions of tea plants will also be the focus of our future study. 

## 4. Materials and Methods

### 4.1. Chemicals

Individual catechins (EGCg, EGC, ECg, EC, GCg, GC, Cg, C, all ≥95%), myricetin (≥98%), quercetin (≥95%) and kaempferol (≥99%) were bought from Sigma-Aldrich (Shanghai, China). Acetonitrile and methanol (HPLC grade) were purchased from Merck KGaA company (Darmstadt, Germany). Ethanol was bought from Sinopharm Chemical Reagent Co., Ltd. (Shanghai, China).

### 4.2. Implementation of Shade Treatments

Five tea cultivars at the same age (<5 years), including ‘Longjing 43’, ‘Zhongming 192’, ‘Wanghai 1’, ‘Jingning 1’ and ‘Zhonghuang 2’, were used for the study (Songyang Tea Plantation of Lishui Academy of Agricultural Sciences, Lishui County, Zhejiang, China, 28°577’ N, 119°377’ E). Two black polyethylene nets (shade percentages of 70% and 95%), a blue polyethylene shade net (shade percentage of 95%), a yellow polyethylene shade net (shade percentage of 90%) and a red polyethylene shade net (shade percentage of 95%) were used to cover each tea cultivar from the end of June to the middle of July in 2020, with a height of 2 m above ground level and a size of 4 m × 6 m. All shade nets were purchased from Taizhou Huiming Shade Net Co., Ltd. (Taizhou, China). Open all-around shade was used to achieve consistent ambient temperatures under different shade treatments. A TOP Instruments light intensity sensor (Zhejiang Top Instrument Co., Ltd., Hangzhou, China) was used to measure the light intensity at the height of the tea shoots on the 13th July of 2020 (the 20th day), while the UV intensity was measured using a TENMARS UVAB light meter (TENMARS Electronics Co., Ltd, Taiwan, China). A HopooColor OHSP-350C illumination analyzer (HopooColor Technology Co., Ltd., Hangzhou, China) was used to record the intensity of the visible spectrum on the same day. 

### 4.3. Preparation of Fixed Tea Samples

Tea leaves for each treatment (around 30 g fresh weight) were harvested from different tea cultivars at the standard of two leaves and one bud after shade treatments of 5, 10, 15, 20 and 25 sunny days. To preserve the authentic chemical compositions of fresh tea leaves, the harvested tea leaves were immediately submitted to pan fixation at 270 °C for 3 min, followed by oven drying at 120 °C for 1 h. All tea samples were stored at 4 °C prior to chemical analyses.

### 4.4. Analysis of Catechins and Flavonol Glycosides in Tea Samples

The tea extract was prepared and the flavonoid composition was analyzed using our reported method [4]. After grinding and sifting through a 0.45 mm sifter, the obtained tea powder (150 mg) was extracted with 25 mL of 50% (*v*/*v*) ethanol solution once in a water bath (HZ-9211 KB water bath shaker) at 70 °C and 100 rpm for 30 min. The mixture was centrifuged at 5000 rpm and 4 °C for 10 min and the supernatants were submitted to UHPLC-DAD-MS/MS analysis. 

All samples were centrifuged again (13,000 rpm, 4 °C, 15 min) and then the supernatants were collected for analysis using an ACQUITY Ultra Performance LC™ instrument equipped with a Waters Quattro Premier XE Mass Spectrometer (Waters Corporation, Milford, MA, USA) according to our previously reported method [4]. Briefly, the UHPLC conditions were: Waters ACQUITY UPLC HSS T3 column (2.1 mm × 150 mm, 1.8 µm), column temperature 35 °C, injection volume 4 µL, mobile phase A = 0.1% formic acid +99.9% water (*v*/*v*), mobile phase B = 0.1% formic acid +99.9% acetonitrile (*v*/*v*), linearly increasing from 95.0% (*v*) A/5.0% (*v*) B to 67.0% (*v*) A/33.0%(*v*) B during the 30 min, then 95.0% (*v*) A/5.0% (*v*) B until 35 min for re-equilibration, flow rate 0.3 mL·min^−1^. An electrospray ionization (ESI) technique in negative ion mode was employed for MS scans using the same MS conditions previously reported for peak identification and supplemental quantification. Catechins were quantified using the authentic standards and flavonol glycosides were relatively quantified by their aglycones [21].

### 4.5. Association of Flavonoid Compounds with the Light Condition Parameters

The coexpression network of flavonoids was constructed using WGCNA R package (v 1.47) [45]. The Pearson correlations among all flavonoids were calculated and the adjacency matrix was created. The soft-thresholding power β was set to 7. Next, the adjacency matrix was used to calculate the topological overlap measure (TOM). The dissimilarity TOM was used as an input for the dendrogram. The modules were established using the DynamicTreeCut algorithm and assigned to different colors [45]. The light and UV intensities were used for correlations with different modules.

### 4.6. Data Analysis

All tests were repeated three times and the results (mean values ±SD) were presented. A significant difference analysis was carried out using the SAS System (Windows version 8.1, SAS Institute Inc., Cary, NC, USA) and Tukey’s test. Principal component analysis (PCA) was conducted using Minitab 17 statistical software (Minitab. LLC, State College, PA, USA) and the co-expression network was constructed using Cytoscape software (version 3.8.0).

## 5. Conclusions

The present study investigated the effects of black net and colored net shade treatments on the flavonoid compositions of the young shoots of ‘Longjing 43’, ‘Zhongming 192’, ‘Wanghai 1’, ‘Jingning 1’ and ‘Zhonghuang 2’ tea cultivars. The different tea cultivars had different compositions of flavonoids, which also showed differential responses to shade treatments. In general, flavonol glycosides were more sensitive to light condition changes than catechins. The contents of TFG in the young shoots of the different tea cultivars generally decreased with the decreases in light and UV intensities. The aglycone moiety of flavonol glycosides played a more important role in regulating the accumulation of flavonol glycosides responding to altered light conditions compared with the sugar moiety. M-glycosides and Q-glycosides showed greater sensitivity to light conditions than K-glycosides. The PCA results showed that the profiles of flavonoids in the different tea samples were modulated by shade treatments based on the corresponding cultivar features. The WGCNA results indicated that the contents of most flavonol glycosides were positively correlated with the intensities of light and UV rather than the blue, green, yellow and red ratios in the light spectrum, while only Q-glu-rha-rha responded to light quality changes. UV is the key region of sunlight promoting the accumulation of flavonol glycosides in tea leaves, along with light quality changes. 

## Figures and Tables

**Figure 1 molecules-26-07158-f001:**
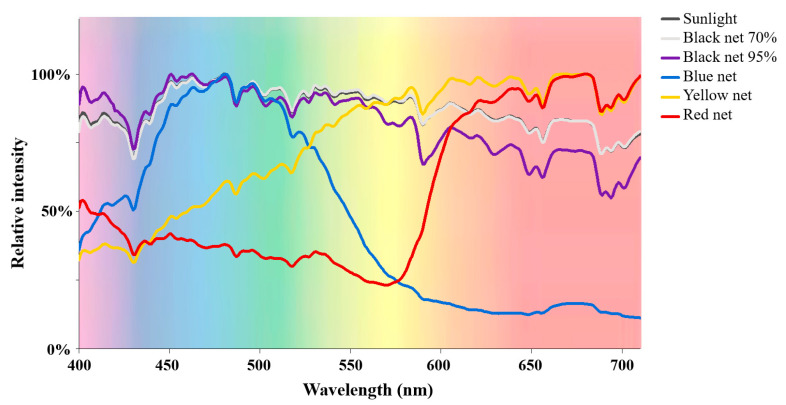
The light spectrograms under different shade treatments.

**Figure 2 molecules-26-07158-f002:**
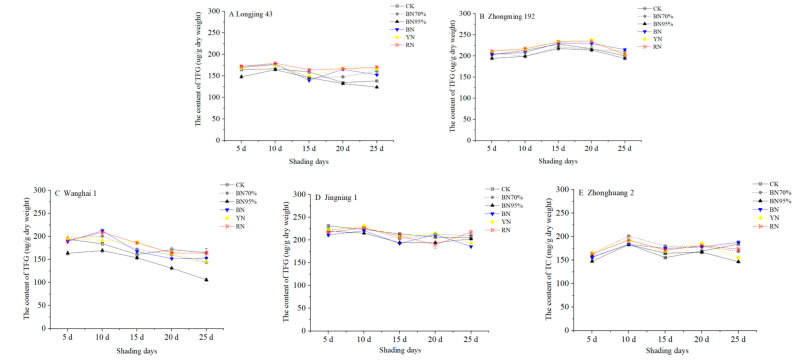
The impacts of shade treatments on the contents of total catechins in the young shoots of different tea cultivars. TC: total catechins. Significantly different results are shown in Appendix A.

**Figure 3 molecules-26-07158-f003:**
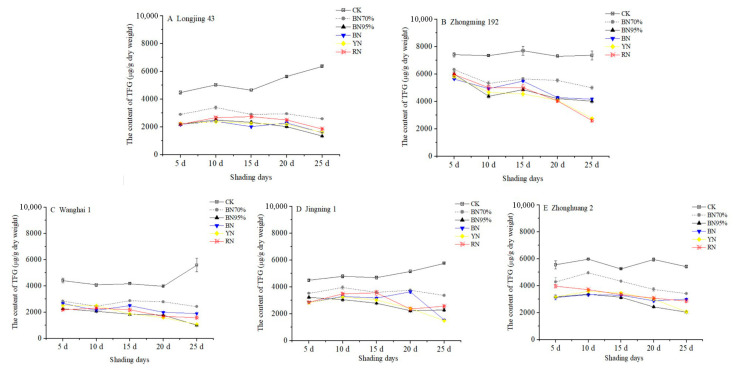
The impacts of shade treatments on the contents of total flavonol glycosides in the young shoots of different tea cultivars. TFG: total flavonol glycosides. Significantly different results are shown in Appendix A.

**Figure 4 molecules-26-07158-f004:**
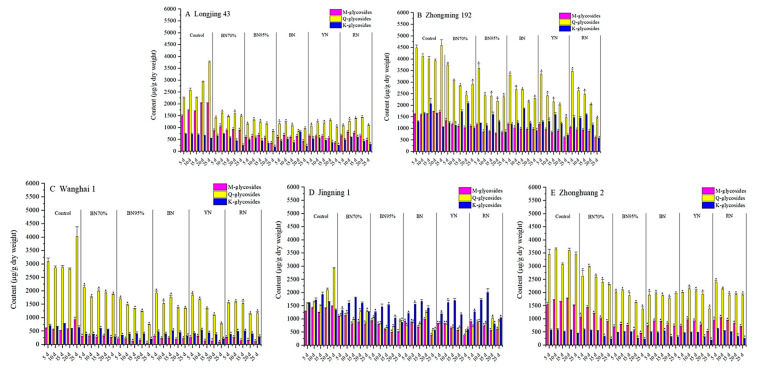
Impacts of different shade treatments on the contents of flavonol glycoside groups classified by aglycone. M-glycosides: myricetin glycosides; Q-glycosides: quercetin glycosides; K-glycosides: kaempferol glycosides. For the same tea cultivars harvested on the same day, data annotated with * were significantly different from the corresponding CK (*p* < 0.05). Significant difference analysis was carried out using the TTEST function of Microsoft Excel for Mac (version 16.54, Microsoft Corporation, Redmond, WA, USA).

**Figure 5 molecules-26-07158-f005:**
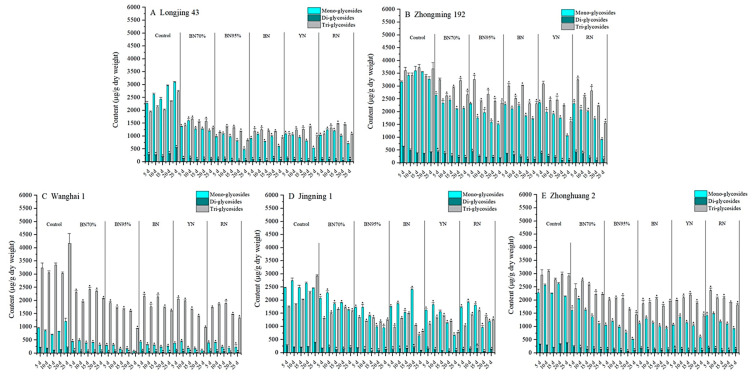
Impacts of different shade treatments on the contents of flavonol glycosides classified by sugar moiety. For the same tea cultivars harvested on the same day, data annotated with * were significantly different from the corresponding CK (*p* < 0.05). Significant difference analysis was carried out using the TTEST function of Microsoft Excel for Mac (version 16.54, Microsoft Corporation, Redmond, WA, USA).

**Figure 6 molecules-26-07158-f006:**
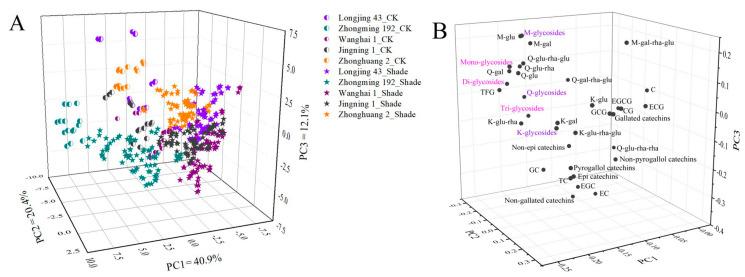
The PCA results for the tea samples of different cultivars under shade treatments based on flavonoid composition: (**A**) score plot; (**B**) loading plot. The number of replicates was equal to 3.

**Figure 7 molecules-26-07158-f007:**
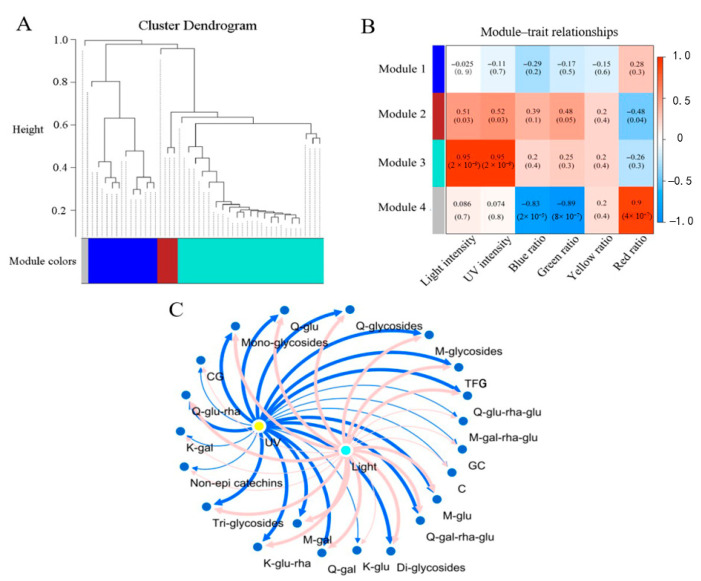
Correlations of flavonoid contents with light parameters via WGCNA. Hierarchical clustering dendrogram (**A**). The modules achieved are shown with correlation coefficients (upper) and *p* values (lower) annotated in each cell (**B**). Correlation network of light intensity (aquamarine blue node), UV intensity (yellow node) and the contents of flavonoids (blue nodes) for module 3 (**C**). The co-expression network construction was performed using Cytoscape software (version 3.8.0). The line width was increased in response to the elevated correlation coefficient from 0.204 to 0.825.

**Table 1 molecules-26-07158-t001:** The basic light condition parameters under different shade nets.

	Sunlight	Black Net 70%	Black Net 95%	Blue Net	Yellow Net	Red Net
Light intensity (lux)	12,686 ± 407	5165 ± 729	892 ± 300	1713 ± 357	1945 ± 196	2938 ± 272
UV intensity (µw/cm^2^)	4823 ± 258	1765 ± 120	211 ± 89	185 ± 45	202 ± 110	640 ± 54
Blue ratio in the light spectrum (%)	31.7 ± 0.3	31.5 ± 0.1	34.5 ± 0.3	50.4 ± 0.9	18.8 ± 0.2	22.5 ± 0.2
Green ratio in the light spectrum (%)	25.8 ± 0.1	25.7 ± 0.1	26.6 ± 0.3	34.5 ± 0.3	24.6 ± 0.1	13.2 ± 0.3
Yellow ratio in the light spectrum (%)	6.4 ± 0.1	6.4 ± 0.1	6.1 ± 0.1	3.1 ± 0.1	7.5 ± 0.1	3.3 ± 0.1
Red ratio in the light spectrum (%)	36.2 ± 0.3	36.4 ± 0.1	32.8 ± 0.1	12.0 ± 0.5	49.1 ± 0.3	61.0 ± 0.6

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
