# Peer review of "Screening the Key Region of Sunlight Regulating the Flavonoid Profiles of Young Shoots in Tea Plants (Camellia sinensis L.) Based on a Field Experiment"

_molecules, 2021, doi:10.3390/molecules26237158_

Round 1

Reviewer 1 Report

This manuscript describes in detail the influence of general shading and shading of different wavelength ranges on the flavanol and flavonol contents in tea leaves. The manuscript is well written and presents detailed data and well discussed results. Specially the fact that also the flavanols were analyzed, is a valuable contribution to knowledge about tea, because they are rarely analyzed although important components.

There are a few details to be corrected or amended:

row 36: antioxidant capacity is not a health benefit, see Pompella et al.: “The use of total antioxidant capacity as surrogate marker for food quality and its effect on health is to be discouraged.,” Nutrition, vol. 30, no. 7, pp. 791–793, Jul. 2014.

Introduction: I suggest to mention also Dörr et al.: “Influence of leaf temperature and blue light on the accumulation of rosmarinic acid and other phenolic compounds in Plectranthus scutellarioides (L.),” Environmental and Experimental Botany, vol. 167, p. 103830, Nov. 2019.

Introduction: it should be mentioned that shading of tea plants is a common practice for producing high grade green tea at least in Japan (gyokuro, kabusecha). I’m not sure if this shading is common also in other countries, but given the importance of Japanese green tea, this kind of shading can’t remain unmentioned.

Figure 1: it is not clear, what is the basic value of the relative intensity.

Figures 2 to 7: these figures must be vector graphics and not bitmaps, in order to allow for zooming in without getting blurry.

rows 140 and 163: it seems that “total catechins” is the sum of the catechin contents as determined by UHPLC. This should be stated clearly, in order to clarify that the total catechins were not measured photometrically. The same must be clarified for the term “total flavonol glycosides”.

row 394: “Hesse” has to be deleted.

row 397: it is not clear, how old the tea plants are. The exact age is not relevant, but at least an approximate indication like “less than five years”, “older than ten years” or whatever is applicable, should be given.

rows 423ff: the description of centrifugation and extraction steps is not clear. Was the supernatant of a single extraction analyzed or were the combined extracts after two extractions analyzed?

row 425: the analytical technique as such is called UHPLC, not UPLC. UPLC is the name of the UHPLC instruments by Waters. When mentioning the analytical technique as in this row, the company name must not be stated.

rows 426ff: change “the UPLC conditions” to “the UHPLC conditions” in row 427. The method is not completely described in the reference [20], because [20] refers to another paper for full method description, namely Zheng et al.: “Screening the cultivar and processing factors based on the flavonoid profiles of dry teas using principal component analysis,” J. Food Compos. Anal., pp. 1–39, Dec. 2017. Therefore, this paper must be referenced here. Anyway, in Zheng et al. (2017), the instrumentation is not described in detail, since there are different Waters UPLC models with various options for pumps, autosamplers, DAD detectors. Also, the model of the mass spectrometer is not defined. These details must be added in this manuscript.

MS data were not used for quantitative evaluation. What was it used for? For confirmation of peak identity? If so, please add this information.

Author Response

This manuscript describes in detail the influence of general shading and shading of different wavelength ranges on the flavanol and flavonol contents in tea leaves. The manuscript is well written and presents detailed data and well discussed results. Specially the fact that also the flavanols were analyzed, is a valuable contribution to knowledge about tea, because they are rarely analyzed although important components.

There are a few details to be corrected or amended:

row 36: antioxidant capacity is not a health benefit, see Pompella et al.: “The use of total antioxidant capacity as surrogate marker for food quality and its effect on health is to be discouraged.,” Nutrition, vol. 30, no. 7, pp. 791–793, Jul. 2014.

Response: ‘antioxidant capacity’ was changed to “anti-tumour’ (New line:37).

Introduction: I suggest to mention also Dörr et al.: “Influence of leaf temperature and blue light on the accumulation of rosmarinic acid and other phenolic compounds in Plectranthus scutellarioides (L.),” Environmental and Experimental Botany, vol. 167, p. 103830, Nov. 2019.

Response: The reference was cited in the text (New lines: 50, 51).

Introduction: it should be mentioned that shading of tea plants is a common practice for producing high grade green tea at least in Japan (gyokuro, kabusecha). I’m not sure if this shading is common also in other countries, but given the importance of Japanese green tea, this kind of shading can’t remain unmentioned.

Response: Thanks for the suggestion. Complied (New lines: 53-55).

Figure 1: it is not clear, what is the basic value of the relative intensity.

Response: The quality of Figure 1 has been improved, and the value of relative intensity has been corrected (using percentages).

Figures 2 to 7: these figures must be vector graphics and not bitmaps, in order to allow for zooming in without getting blurry.

Response: The quality of Figures 2-7 has been improved.

rows 140 and 163: it seems that “total catechins” is the sum of the catechin contents as determined by UHPLC. This should be stated clearly, in order to clarify that the total catechins were not measured photometrically. The same must be clarified for the term “total flavonol glycosides”.

Response: The definition of TC and TFG has been clarified in the text (New lines: 126, 149) . No colorimetric method is used in our study, thus the contents of TC and TFG should be the sums of the corresponding individual compounds.

row 394: “Hesse” has to be deleted.

Response: “Hesse” has been deleted.

row 397: it is not clear, how old the tea plants are. The exact age is not relevant, but at least an approximate indication like “less than five years”, “older than ten years” or whatever is applicable, should be given.

Response: The age of these tea plants is four years old, which has been indicated in the text (New line: 384).

rows 423: the description of centrifugation and extraction steps is not clear. Was the supernatant of a single extraction analyzed or were the combined extracts after two extractions analyzed?

Response: The supernatant of a single extraction was analyzed, which was indicated in the text (New line: 410). Based on our previous study, a single extraction adequately extracted the flavonoids from tea leaves, and there is no need of two extractions (preliminary experiment). Our method has been previously published (Screening the cultivar and processing factors based on the flavonoid profiles of dry teas using principal component analysis, Journal of Food Composition and Analysis, 2018, 67, 29–37).

row 425: the analytical technique as such is called UHPLC, not UPLC. UPLC is the name of the UHPLC instruments by Waters. When mentioning the analytical technique as in this row, the company name must not be stated.

Response: “UPLC” was changed to “UHPLC”. This part was rewritten accordingly (New lines: 415-417).

rows 426: change “the UPLC conditions” to “the UHPLC conditions” in row 427. The method is not completely described in the reference [20], because [20] refers to another paper for full method description, namely Zheng et al.: “Screening the cultivar and processing factors based on the flavonoid profiles of dry teas using principal component analysis,” J. Food Compos. Anal., pp. 1–39, Dec. 2017. Therefore, this paper must be referenced here. Anyway, in Zheng et al. (2017), the instrumentation is not described in detail, since there are different Waters UPLC models with various options for pumps, autosamplers, DAD detectors. Also, the model of the mass spectrometer is not defined. These details must be added in this manuscript.

Response: The“UPLC” was changed to “UHPLC” (New line: 413). Actually, “Screening the cultivar and processing factors based on the flavonoid profiles of dry teas using principal component analysis,” J. Food Compos. Anal., pp. 1–39 is also our published work, which was cited instead (New line: 417). The instrument used for flavonoid analysis is the same, viz. ACQUITY Ultra Performance LC™ equipped with Waters Quattro Premier XE Mass Spectrometer (Waters Corporation, Milford, MA, USA, which has been indicated in the text (New lines: 415-417).

MS data were not used for quantitative evaluation. What was it used for? For confirmation of peak identity? If so, please add this information.

Response: MS data is used for both peak identification and supplemental quantification of certain compounds like C (described in our previous work Screening the cultivar and processing factors based on the flavonoid profiles of dry teas using principal component analysis,” J. Food Compos. Anal., pp. 1–39). This has been indicated in the text (New lines: 424, 425).

Reviewer 2 Report

Authors investigated influence of wavelength for flavonoid profile in several cultivar of tea plant. As mentioned in their explanation, influence of wavelength and difference cultivar for productivity of flavonoids has been already many researchers reported. Although information is not new, if there is a meaningful point of this report, it is that their experiment had been performed in the field by sunlight, not LED light. Authors show the huge data; however, it is hard to follow because of long explanation. I recommend drastically rewrite to make it shorter. Authors investigated influence of wavelength for flavonoid profile in several cultivar of tea plant. As mentioned in their explanation, influence of wavelength and difference cultivar for productivity of flavonoids has been already many researchers reported. Although information is not new, if there is a meaningful point of this report, it is that their experiment had been performed in the field by sunlight, not LED light. Authors show the huge data; however, it is hard to follow because of long explanation. I recommend drastically rewrite to make it shorter.

L20,21: What is M-glycosides, Q-glycosides and K-glycosides?

Introduction section is too heavy. Should be simplified L58-72: too much information. L86-92: should be removed. In section 2.2 is hard to read. Many data were described in this section. It is hard to follow because author discussed together difference of cultivar and treatment. Figure 3 and 4, no standard errors and statistical analysis results were provided.

“L206 By contrast,.” and “L207 By contrast, …”, they are same contents.

Explanation of "Result" and "discussion" section has overlapped each other. I recommend combining the two into one as “result and discussion” section.

Author Response

Authors investigated influence of wavelength for flavonoid profile in several cultivar of tea plant. As mentioned in their explanation, influence of wavelength and difference cultivar for productivity of flavonoids has been already many researchers reported. Although information is not new, if there is a meaningful point of this report, it is that their experiment had been performed in the field by sunlight, not LED light. Authors show the huge data; however, it is hard to follow because of long explanation. I recommend drastically rewrite to make it shorter.

Response: We admit the influence of wavelength and difference cultivars for productivity of flavonoids has been reported, however many studies only select one or two tea cultivars for investigation. Here are the examples of references: Regulation of growth and flavonoid formation of tea plants (Camellia sinensis) by blue and green light. J Agr Food Chem 2019, 67, (8), 2408-2419; Exploration of the effects of different blue LED light intensities on flavonoid and lipid metabolism in tea plants via transcriptomics and metabolomics. Int J Mol Sci 2020, 21, (13); Metabolic flux redirection and transcriptomic reprogramming in the albino tea cultivar 'Yu-Jin-Xiang' with an emphasis on catechin production. Sci Rep 2017, 7, 45062; Metabolomics analysis reveals the metabolic and functional roles of flavonoids in light-sensitive tea leaves. BMC plant biology 2017, 17; Transcriptomic analysis reveals mechanism of light-sensitive albinism in tea plant Camellia sinensis 'Huangjinju'. BMC plant biology 2020, 20, (1); Shading effects on leaf color conversion and biosynthesis of the major secondary metabolites in the albino tea cultivar "Yujinxiang". J Agric Food Chem 2020, 68, (8), 2528-2538. It is still difficult to compared the chemical results between different studies due to the different analysis methods or different standards used. Thus, a wider comparison between different tea cultivars is needed to have a clearer concept about the response of different tea cultivars to light condition regarding flavonoid biosynthesis. Besides, as mentioned in the manuscript, the LED facilities are difficult for wide application in the field, due to the limited intensity of LED, the supply of electric power and the high cost for long-term service and maintenance (New lines 72-74). Most studies were carried out under artificial light condition, and the respective contributions of UV, blue and red light regions of sunlight to the biosynthesis of flavonoids are still unclear (New lines 67-69). We consider it is necessary to understand the contributions of the key region of sunlight to the biosynthetic regulation of flavonoids based on a field experiment for not only practical production instruction but also future study direction about the light-induced regulation on flavonoids in tea plants, focusing on UVB or light quality? Nevertheless, the manuscript has been condensed.

L20,21: What is M-glycosides, Q-glycosides and K-glycosides? 

Response: The abbreviations were deleted, and the full names (Myricetin glycosides, quercetin glycosides, kaempferol glycosides) were given in the abstract (New lines: 22, 23).

Introduction section is too heavy. Should be simplified L58-72: too much information. L86-92: should be removed. In section 2.2 is hard to read. Many data were described in this section. It is hard to follow because author discussed together difference of cultivar and treatment. Figure 3 and 4, no standard errors and statistical analysis results were provided.

Response: Old lines 58-72 have been condensed (New lines:56-67). Old lines 86-92 have been removed. Section 2.2 has been improved and condensed. There are error bars in Figure 3 and 4, and statistical analysis results were added to Figure 3 and 4.

“L206 By contrast,.” and “L207 By contrast, …”, they are same contents.

Response: The second “By contrast” was deleted (New line: 189).

Explanation of "Result" and "discussion" section has overlapped each other. I recommend combining the two into one as “result and discussion” section. 

Response: According to the official templet of molecules, the Discussion Section was separated from the Results Section, thus the independent Discussion Section was preferred. We moved the discussion sentences to the Discussion Section (New lines: 340-347).

Reviewer 3 Report

The authors have to suggest the possible reason for much reduction of the concentrations of quercetin - and myricetin-glycosides on reduction of light  and UV-B intensities by use of nets in tea farm lands. Moreover, the roles of the enzymes UGT78DI and UGT73C6 in their biosynthesis in presence of UV-B light.

Author Response

The authors have to suggest the possible reason for much reduction of the concentrations of quercetin - and myricetin-glycosides on reduction of light and UV-B intensities by use of nets in tea farm lands. Moreover, the roles of the enzymes UGT78DI and UGT73C6 in their biosynthesis in presence of UV-B light.

Response: The explanations for the greater variations of M-glycosides and Q-glycosides in response to light condition were added to the text (New lines: 291-297). The roles of the enzymes UGT78DI and UGT73C6 in their biosynthesis in presence of UV-B light were discussed in the text (New lines: 302-304).

Round 2

Reviewer 2 Report

 no comment